# MicroRNAs in Testicular Germ Cell Tumors: The Teratoma Challenge

**DOI:** 10.3390/ijms25042156

**Published:** 2024-02-10

**Authors:** Nuphat Yodkhunnatham, Kshitij Pandit, Dhruv Puri, Kit L. Yuen, Aditya Bagrodia

**Affiliations:** 1Department of Urology, University of California San Diego School of Medicine, La Jolla, CA 92093, USA; nyodkhunnatham@health.ucsd.edu (N.Y.); kpandit@health.ucsd.edu (K.P.); dpuri@health.ucsd.edu (D.P.); k3yuen@health.ucsd.edu (K.L.Y.); 2Department of Urology, University of Texas Southwestern Medical Center, Dallas, TX 75390, USA

**Keywords:** microRNA, testicular cancer, testicular germ cell tumor

## Abstract

Testicular germ cell tumors (TGCTs) are relatively common in young men, making accurate diagnosis and prognosis assessment essential. MicroRNAs (miRNAs), including microRNA-371a-3p (miR-371a-3p), have shown promise as biomarkers for TGCTs. This review discusses the recent advancements in the use of miRNA biomarkers in TGCTs, with a focus on the challenges surrounding the noninvasive detection of teratomas. Circulating miR-371a-3p, which is expressed in undifferentiated TGCTs but not in teratomas, is a promising biomarker for TGCTs. Its detection in serum, plasma, and, potentially, cystic fluid could be useful for TGCT diagnosis, surveillance, and monitoring of therapeutic response. Other miRNAs, such as miR-375-3p and miR-375-5p, have been investigated to differentiate between TGCT subtypes (teratoma, necrosis/fibrosis, and viable tumors), which can aid in treatment decisions. However, a reliable marker for teratoma has yet to be identified. The clinical applications of miRNA biomarkers could spare patients from unnecessary surgeries and allow for more personalized therapeutic approaches. Particularly in patients with residual masses larger than 1 cm following chemotherapy, it is critical to differentiate between viable tumors, teratomas, and necrosis/fibrosis. Teratomas, which mimic somatic tissues, present a challenge in differentiation and require a comprehensive diagnostic approach. The combination of miR-371 and miR-375 shows potential in enhancing diagnostic precision, aiding in distinguishing between teratomas, viable tumors, and necrosis. The implementation of miRNA biomarkers in TGCT care could improve patient outcomes, reduce overtreatment, and facilitate personalized therapeutic strategies. However, a reliable marker for teratoma is still lacking. Future research should focus on the clinical validation and standardization of these biomarkers to fully realize their potential.

## 1. Introduction

Testicular germ cell tumors (TGCTs) are the most common malignancy among young men aged 20 to 40 years [1]. According to data from the United States, the incidence of TGCTs is estimated to be 5.55 per 100,000 person-years [2]. There are two subgroups of TGCTs: pure seminomas and non-seminomas, according to the 2022 World Health Organization (WHO) classification. The survival rates for testicular cancer vary based on several factors. These include the stage of cancer and a person’s age and general health. The overall 5-year survival rate for TGCTs is 95%.

There are numerous theories and studies on tumor development using mouse models. Research by Guida E. et al. indicates that the activation of the MAPK pathway in fetal germ cells can lead to the development of aggressive and metastatic teratocarcinomas in both male and female mice. This suggests that the overactivation of MAPK can expand their proliferative window, leading to neoplastic transformation [3]. In a separate study, Pierpont T. M. et al. demonstrated that chemotherapy can selectively eliminate OCT4-positive cancer stem cells in a mouse model of malignant testicular cancer. They concluded that the chemosensitivity of TGCTs derives from the sensitivity of their cancer stem cells to DNA-damaging chemotherapy [4]. Furthermore, research by Meng X. et al. showed that overexpression of the glial cell line-derived neurotrophic factor (GDNF) in undifferentiated spermatogonia promotes the development of invasive malignant testicular tumors containing aneuploid cells [5]. Collectively, these studies highlight the role of specific genetic pathways and factors in the development of testicular tumors and their response to chemotherapy, providing valuable insights into the mechanisms of tumor development and potential therapeutic strategies.

Teratomas, a subset of non-seminoma, are tumors characterized by the presence of well or incompletely differentiated elements from at least two of the three germ cell layers: endoderm, mesoderm, and ectoderm. Notably, these components are intermixed within the tumor. When the tumors exhibit well-differentiated elements, they are classified as mature teratomas, while those with an incomplete differentiation, resembling fetal or embryonal tissue, are termed immature teratomas. Teratomas are typically associated with normal serum tumor markers, although they may lead to a mild elevation in serum AFP levels [6]. Despite being histologically benign, teratomas are genetically unstable, resulting in unpredictable biological behavior. It is important to note that teratomas are resistant to chemotherapy. Although uncommon, these tumors have the potential for rapid growth or the transformation of their ectodermal, mesodermal, and/or endodermal elements, giving rise to a non-germ cell tumor somatic malignancy [7,8,9].

The treatment approach for TGCTs varies depending on histological subtypes and the disease stage, as outlined in the NCCN Clinical Practice Guidelines in Oncology (NCCN Guidelines^®^) version 1.2023. Distinct treatment strategies are employed for seminomas and non-seminomas. The main difference between these two subtypes is the absence of a role for radiation therapy in non-seminomas. For non-seminomas, in Stage 1, localized to the testicle, surveillance with regular check-ups is common due to the high curability of the early-stage disease. Moving to Stage II, where cancer has spread to regional lymph nodes, treatment options may include retroperitoneal lymph node dissection (RPLND), radiation therapy, or chemotherapy. In Stage III, with cancer in more distant lymph nodes or organs, a more aggressive approach is often required. Treatment may include chemotherapy, RPLND, and metastasectomy, with the decision being based on individual patient factors and disease extent.

In current clinical practice, surgical management with post-chemotherapy retroperitoneal lymph node dissection (pcRPLND) is necessary in patients with residual masses ≥ 1 cm. A significant proportion of such cases, approximately 40–45%, harbor teratomas, renowned for their chemotherapy resistance, and subsequently demand surgical excision; 10–15% have residual germ cell tumors that may be effectively treated with surgery alone or additional chemotherapy cycles; and up to 40% of patients may have necrosis only [10,11,12]. While surgical intervention is not necessary for patients exhibiting necrosis, the current limitations in our diagnostic capabilities hinder the precise differentiation between fibrosis/necrosis and the presence of teratoma or viable germ cell tumor (GCT) elements. The implementation of pcRPLND procedures, although valuable, is not without its inherent challenges. Notably, surgeons may encounter desmoplastic reactions within the retroperitoneal cavity, adding a layer of complexity to the process. In certain scenarios, the intricate nature of the surgery may demand supplementary procedures such as nephrectomy or vascular reconstruction to address specific challenges arising during the intervention. However, it is imperative to acknowledge that these additional measures can potentially lead to severe complications, including retroperitoneal hemorrhage or the development of chylous ascites. Therefore, the decision to pursue surgical management should be carefully weighed against the potential risks and complications associated with the specific clinical context, emphasizing the need for a thorough and individualized approach to patient care. Consequently, the prevailing biomarkers exhibit inadequacies in distinguishing individuals necessitating surgical interventions for post-chemotherapy residual masses, thus accentuating the urgency for the development of a diagnostic test capable of precisely identifying teratoma. Furthermore, these conventional markers demonstrate ineffectiveness in identifying active germ cell tumors [13].

The limitation of conventional biomarkers, including Alpha-Fetoprotein (AFP), Beta-Human Chorionic Gonadotropin (β-HCG), and Lactate Dehydrogenase (LDH), lies in their restricted sensitivity and specificity, including for the identification of teratoma [14]. In pure seminomas, β-hCG and LDH are elevated in only 28% and 29% of patients, respectively, while AFP is consistently negative. In non-seminomatous germ cell tumors (NSGCT), elevated concentrations of β-hCG, AFP, and LDH are observed in 53%, 60%, and 39% of cases, respectively [15]. LDH predominantly functions as an indicator of cell death, leading to its elevation in the serum across various conditions characterized by a high cell turnover. Moreover, AFP and β-hCG production can be instigated by ailments unrelated to TGCT, potentially influenced by disruptions in endocrine and metabolic processes [14,16].

In recent years, microRNAs (miRNAs) have emerged as compelling candidates for serving as biomarkers in the context of TGCTs [17]. Among the multitude of miRNAs under investigation, the focus has prominently centered on microRNA-371a-3p (miR-371a-3p) and microRNA-375 (miR-375). The significance of these specific miRNAs lies in their potential to revolutionize our diagnostic and prognostic approaches to TGCTs. This comprehensive literature review critically evaluates recent studies exploring the potential of miRNA biomarkers in TGCTs and their implications for enhancing teratoma identification.

## 2. What Are microRNAs?

MicroRNAs (miRNAs) are small non-coding RNA molecules that play an important role in the regulation of gene expression across various organisms. Typically composed of approximately 18 to 25 nucleotides, they are involved in post-transcriptional gene silencing [18,19]. miRNAs typically exert their regulatory influence by downregulating gene expression through specific interactions with messenger RNAs (mRNAs). This interaction can result in the degradation of the mRNA or the inhibition of mRNA translation, and the outcome depends on the degree of complementarity between the miRNA and its target sequence. The abnormal expression of miRNAs has emerged as a key factor in the development of various human diseases.

The process of miRNA-mediated gene regulation involves the binding of miRNAs to their selectively chosen target mRNAs. Upon binding, these miRNAs can instigate either the degradation of the mRNA or the hindrance of mRNA translation, a regulatory mechanism vital for maintaining cellular homeostasis. Disruptions in this delicate balance, often attributed to aberrant miRNA expression, have been implicated in the etiology of numerous human diseases. Understanding the interplay between miRNAs and gene expression offers valuable insights into the molecular mechanisms underlying pathological conditions, paving the way for potential therapeutic interventions aimed at restoring proper regulatory control in the context of diseases [20,21].

## 3. MicroRNA Signatures as Biomarkers

The concept of an ideal biomarker encompasses a multifaceted criterion crucial for effective clinical application. Firstly, it should be uniquely produced by the malignancy itself, ensuring specificity. Secondly, its secretion into bodily fluids should facilitate reproducible measurements, thereby enhancing diagnostic precision. Furthermore, a crucial aspect is a strong correlation with the tumor’s quantity, allowing for accurate assessment of disease progression. Equally significant is the early detectability of the marker, enabling timely intervention. Additionally, a short half-life is imperative to prevent undue accumulation in body tissues, minimizing the risk of false-positive results. Lastly, the biomarker should exhibit a correlation with the tumor’s response to treatment, offering insights into therapeutic efficacy. This comprehensive set of criteria underscores the complexity and nuance required for an ideal tumor marker, emphasizing its pivotal role in advancing cancer diagnosis and management [22].

There is evidence that suggests that TGCTs imitate embryonic development to a certain extent, retaining the molecular and biochemical characteristics of embryonic stem cells (ESCs). ESCs are a type of stem cell known for their continuous growth, pluripotency, and self-renewal capacity [23]. Notably, ESCs express specific clusters of miRNAs, particularly miR-371-3 and miR-302/367, which play important roles in pluripotency and differentiation. These miRNAs are human ESC-specific, and their expression decreases as cellular development progresses [24,25]. These embryonic miRNAs show promise as potential biomarkers for detecting TGCTs because of their demonstrated stability in serum [26]. Murray et al. were the first to detect measurable levels of these miRNAs in serum. Their research builds upon prior observations that members of the miR-371-373 and miR-302 clusters are consistently overexpressed in all malignant GCTs, regardless of factors like age, subtype, or location. This unique characteristic, not often seen in other cancer types, makes these miRNAs prime candidates for biomarker development. The study focused on a young boy diagnosed with yolk sac tumor. Notably, the serum levels of all eight major members of the two miRNA clusters were found to be elevated at the time of diagnosis. This initial finding reinforces the potential of these miRNAs as reliable indicators of GCT presence. The study also highlights the potential of these miRNAs for disease monitoring. Their consistent overexpression in malignant GCTs suggests that they could be tracked to monitor treatment response and detect possible relapse early on. This could significantly improve patient care and outcomes [27]. Since then, these embryonic miRNAs have emerged as candidates for the next generation of tumor markers for TGCTs.

This article focuses on the roles of miR-371 and miR-375 in various biological processes. miR-371 has been found to be overexpressed in TGCTs, regardless of age, location, or subtype [28]. Targeting miR-371 in these tumors has demonstrated to inhibit growth through cell cycle disruption [28]. Moreover, miR-371 has been identified to suppress the initiation of colon cancer and metastatic colonization by inhibiting the TGFBR2–ID1 signaling axis [29].

In the largest study conducted by Dieckmann et al. [30], a large-scale, multicenter investigation aimed at validating the M371 test as a reliable biomarker for GCTs, researchers compared serum miR-371a-3p levels in 616 GCT patients and 258 healthy controls. They employed quantitative polymerase chain reaction, a highly sensitive technique, to measure miR-371a-3p concentrations. The M371a-3p levels were significantly elevated in the GCT patients compared to the controls, showcasing excellent sensitivity for tumor detection. The researchers also explored the M371 test’s utility in monitoring treatment response and disease recurrence. The miR-371a-3p levels declined significantly after surgery in patients with localized disease, highlighting its potential as a marker for treatment efficacy. In relapsed patients, elevated levels again emerged, demonstrating its potential for the early detection of recurrence and guiding follow-up strategies. The study reported a sensitivity of 91.8%, a specificity of 96.1%, a positive predictive value (PPV) of 97.2%, a negative predictive value (NPV) of 82.7%, and an area under the curve (AUC) of 0.97 [30].

Similarly, Nappi et al. conducted a study involving 110 patients, reporting an impressive sensitivity of 96%, a specificity of 100%, a PPV of 100%, an NPV of 98%, and an AUC of 0.97. Badia et al.’s study, with a cohort of 69 patients, showcased a sensitivity of 93.1%, a specificity of 100%, a PPV of 100%, an NPV of 73.3%, and an AUC of 0.98. Lastly, Sequeira et al. conducted a study with 82 patients, demonstrating a sensitivity of 93.6%, a specificity of 100%, a PPV of 100%, an NPV of 96%, and an AUC of 0.98. Collectively, these studies highlight the promising diagnostic accuracy of the evaluated tests, offering valuable insights into their performance metrics across different cohorts and contributing to the evolving landscape of TGCT diagnostics [31,32,33] (Table 1).

On the other hand, miR-375 is a microRNA that serves as a multifunctional regulator in a wide array of physiological and pathological cellular processes. It plays a key role in the differentiation and functioning of cells within the nervous and immune systems, bone, and adipose tissue and even impacts the life cycle of several viruses [34]. In addition, miR-375 has a significant role in diabetes management as it regulates the expression of genes involved in the formation of pancreatic islets, pancreatic development, and β-cell secretion [35]. miR-375 also facilitates the neurogenesis of spinal motor neurons by targeting the cyclin kinase CCND2 and transcription factor PAX6. It acts as an inhibitor for tumor suppressor p53, thereby protecting neurons from apoptosis in response to DNA damage [36].

In the field of oncology, miR-375 has emerged as a pivotal microRNA with potential implications in the detection of testicular cancer. Recent genomic and epigenomic analyses of testicular tumors have revealed a noteworthy upregulation of miR-375, particularly within teratomas. This increased expression suggests a potential association between miR-375 and teratomas, thereby accentuating its candidacy as a discerning biomarker for testicular cancer. The unique molecular signature of miR-375 within teratomas underscores its potential contribution to the diagnostic landscape, offering insights which could enhance our ability to identify and differentiate subtypes of testicular cancer [37].

## 4. Differentiating Teratoma

Teratomas, intriguingly, frequently exhibit an elevated level of cellular differentiation, showcasing a remarkable resemblance to typical somatic tissues. This inherent complexity introduces a significant hurdle in the quest for an appropriate circulating biomarker. The intricacy arises due to the fact that the plasma compartment within the bloodstream functions as a reservoir for an extensive array of molecules released from diverse cells and tissues distributed throughout the entirety of the human body [6].

The challenge is further compounded by the nuanced nature of teratomas, which, in their differentiated state, often mirror the appearance and characteristics of normal tissues [6]. This shared phenotypic similarity complicates the identification of specific biomarkers that can reliably distinguish teratomas from surrounding healthy tissues.

A notable characteristic of miR-371 that sets it apart from contemporary biomarkers is its ability to show heightened levels in the serum of individuals affected by both seminoma and non-seminoma cases. This distinctive attribute positions miR-371 as a promising diagnostic tool, offering a broad-spectrum indicator for various testicular cancer types. However, it is crucial to emphasize that miR-371, while proficient in detecting the presence of testicular cancer, lacks the ability to discriminate between specific teratoma subtypes. This limitation underscores the necessity for a comprehensive diagnostic approach, acknowledging the diversity within the teratoma spectrum and prompting further exploration for complementary biomarkers that can provide a more refined and accurate characterization of distinct teratoma subtypes.

To investigate the utility of miR-371 in teratoma detection, Dieckmann et al. conducted a study considering miR-371a-3p expression within the cystic fluid derived from post-chemotherapy teratomas. The study cohort consisted of four patients with residual cystic teratomas lacking vital undifferentiated GCT tissue, all of whom had undergone RPLND. The analytical evaluation of cystic fluid revealed elevated miR-371 levels in three of these patients, with traces of miR-371 observed in one patient [38].

Several studies have investigated miR-375’s potential role in the context of teratoma. Myeklebust et al. found that miR-375-3p displayed the highest expression levels within the teratoma group. However, when compared to the control group, no significant differences in blood levels were discerned. On the other hand, miR-222-5p, miR-200a-5p, miR-196b-3p, and miR-454-5p were found to be ineffective in distinguishing teratoma from other conditions [39].

In an insightful investigation conducted by Lafin et al. [40], the study delved into the potential utility of miR-375 as a discerning marker for teratoma. This forward-looking research meticulously recruited 40 patients already diagnosed with TGCT and scheduled for pcRPLND. The researchers collected preoperative serum samples from the participants to meticulously assess the expression levels of miR-375-3p and miR-375-5p. The obtained results unveiled intriguing findings, with miR-375-3p showcasing a sensitivity of 86% and a specificity of 32%, resulting in an area under the curve (AUC) of 0.506. On the other hand, miR-375-5p exhibited a sensitivity of 55% and a specificity of 67%, with an AUC of 0.556 [40]. Additionally, Moore et al.’s study uncovered a potential link between teratoma and the expression levels of specific serum miRNAs: miR-375, miR-200a-3p, miR-200a-5p, and miR-200b-3p. This investigation involved 22 patients diagnosed with NSGCT who exhibited residual NSGCT after chemotherapy and subsequently underwent post-chemotherapy consolidation surgery. The cohort was divided into two groups: an 11-member teratoma group and an 11-member necrosis/fibrosis/viable tumor group. The findings of this study did not reveal a statistically significant association between the expression levels of the aforementioned miRNAs and the presence of teratoma. Specifically, the AUC for miR-375 was calculated to be 0.62, indicating a lack of significant discriminatory power for this miRNA in distinguishing teratoma [41].

Lastly, Belge et al. conducted a study investigating miR-375-3p as a potential biomarker for teratoma. The study enrolled 21 TGCT patients with teratoma, 12 patients with other TGCTs, and 12 male controls. The findings of the study indicated that there were no significant differences in the serum miR-375-3p levels among the teratoma patients, other TGCT patients, or the control group, as evidenced by an AUC of 0.524. Consequently, the study suggests that miR-375-3p may not be a suitable biomarker for identifying teratoma [42] (Table 2).

In addition to miRNAs, researchers are investigating other potential biomarkers, such as genetic markers, proteins, and other molecules. The roles of the High Mobility Group A (HMGA) protein family, PATZ1, and the G Protein-Coupled Estrogen Receptor (GPR30) in TGCTs and spermatogenesis are of particular interest. HMGA proteins, including HMGA1a, HMGA1b, and HMGA2, are chromatin architectural factors that participate in the transcriptional regulation of numerous genes [43,44]. They are strongly expressed during embryogenesis and have been found to be upregulated in several human tumors. These proteins are expressed in different stages of spermatogenesis and could serve as a diagnostic tool in the event of a controversial TGCT diagnosis [45]. PATZ1, a transcriptional repressor, plays a key role in spermatogenesis. Its tumor suppressor activity has been found to be impaired in TGCT due to its delocalization into the cytoplasm of cancer cells. This is likely due to the downregulation of the estrogen receptor β, a molecular partner which interacts with PATZ1 [46,47]. Lastly, estrogens play key roles in testis physiology. Their receptors, ERα and ERβ, are strongly expressed in specific testis cells. The role of other estrogen molecules in testis physiology is currently under investigation [48].

## 5. Combination of miRNAs

Two interesting studies explored the combination of miR-371 and miR-375 for distinguishing teratoma, viable tumor, and necrosis. Essentially, researchers wanted to see if these two specific molecular markers, when used together, could improve our ability to distinguish between different health conditions. By looking at how miR-371 and miR-375 work together, the studies aimed to enhance the accuracy of diagnostic assessments, especially in situations where it is crucial to differentiate between teratoma, viable tumor, and necrosis. Using both miR-371 and miR-375 together might create a sort of “molecular fingerprint”, offering a more detailed view which could improve our ability to diagnose and understand different tissue states.

First, Nappi et al. investigated the accuracy of plasma miR-375 alone or in combination with miR-371 in detecting teratomas. This prospective multi-institutional study included 100 GCT patients divided into two cohorts: a discovery cohort and a validation cohort. In the discovery cohort, 20 patients with pure teratoma and 42 patients with no/low risk of harboring teratoma were compared. The validation cohort included 21 patients with confirmed teratoma, 6 patients with active germ cell malignancy, and 11 patients with a complete response after chemotherapy. The results showed that the AUC values for miR-375, miR-371, and miR-371-miR-375 were 0.93, 0.59, and 0.95, respectively, in the discovery cohort. In the validation cohort, the AUC values were 0.55, 0.74, and 0.77 for miR-375, miR-371, and miR-371-miR-375, respectively. The study concluded that plasma miR-371-miR-375 integrated evaluation is highly accurate in detecting teratoma. However, the study had limitations, including the relatively small number of teratoma patients and the lack of postoperative blood tests [49].

Secondly, Kremer et al. studied the potential of combining miR-371a-3p and miR-375-5p to differentiate between viable GCTs and teratomas from necrosis in pcRPLND specimens. The study included 48 metastatic GCT patients and was stratified into three groups—viable GCT (*n* = 16), pure teratoma (*n* = 16), and necrosis/fibrosis (*n* = 16)—all within the pcRPLND setting. The article revealed that, by combining miR-371a-3p and miR-375-5p in pcRPLND tissue samples, they were able to distinguish viable GCTs and teratomas from necrosis/fibrosis with a sensitivity of 93.8%, a specificity of 93.8%, a PPV of 96.8%, an NPV of 88.2%, and an AUC of 0.938. This combination of miRNAs could serve as a new biomarker in the future, potentially sparing miRNA-negative patients from pcRPLND and reducing overtreatment. Nevertheless, the study had limitations, including a retrospective design and the absence of external validation of the findings on patient serum [50].

Despite limitations in both studies, such as small sample sizes and retrospective designs, the combined use of miR-371 and miR-375 presents a promising step forward in enhancing diagnostic precision and sparing patients from unnecessary overtreatment. Further research and validation efforts are warranted to solidify the utility of this molecular approach in clinical practice.

## 6. Future Direction

### 6.1. Combination with Messenger RNA (mRNA)

A recent study published in Cancer 2023 offers a beacon of hope, shedding light on a potential solution to this longstanding dilemma [51]. Researchers embarked on a multifaceted investigation, meticulously dissecting the molecular landscape of post-chemotherapy lymph node tissue. Their approach involved meticulously classifying 48 patients into three groups: those with teratoma, those with viable GCT, and those with necrosis. Using a technique known as microdissection, they precisely isolated representative areas of each tissue type within the lymph nodes. This granular approach allowed them to delve deeper into the molecular signatures of each entity, paving the way for a more nuanced understanding.

Their analysis, employing advanced technologies like NanoString and proteomics, focused on both the blueprint for protein production (messenger RNA or mRNA) and the actual protein levels present in each tissue type. This multi-omics approach yielded striking results: two key players emerged—anterior gradient protein 2 homolog (AGR2) and keratin, type I cytoskeletal 19 (KRT19). Both proteins were found to be significantly overexpressed in teratoma compared to necrosis, both at the mRNA and protein levels. This differential expression pattern offered a tantalizing clue, a potential molecular fingerprint which could distinguish the two entities with unprecedented accuracy.

Further validation using immunohistochemistry (IHC) in both the initial cohort and an independent group of patients solidified the findings. This robust validation step underscored the potential of AGR2 and KRT19 as reliable biomarkers for differentiating teratoma from necrosis in clinical practice. The implications of this breakthrough are manifold. Firstly, it opens the door to pre-surgical diagnostics, potentially sparing patients with confirmed necrosis from unnecessary and potentially debilitating procedures like pcRPLND. Secondly, it paves the way for the development of targeted therapies specifically aimed at teratoma, exploiting the unique vulnerabilities exposed by AGR2 and KRT19 overexpression. Finally, a more accurate diagnosis empowers oncologists to craft personalized treatment plans, tailoring interventions to the specific needs of each patient and potentially improving the overall outcomes and quality of life [51].

While further research is needed to translate these promising findings into routine clinical practice, the aforementioned study represents a significant leap forward in the quest for precise diagnosis and personalized care in testicular cancer patients. The potential arises to develop a novel tool for distinguishing teratoma by combining it with the use of miRNA.

### 6.2. Artificial Intelligence Technology

Artificial intelligence (AI) is rapidly permeating this field, offering unprecedented tools for diagnosis, treatment, and patient care. An iTRUE study delves deep into the current applications of AI in urology and paints a captivating picture of its future potential.

One of the most promising applications of AI lies in enhanced diagnostics. Imagine a world where complex imaging like CT scans and MRIs are analyzed not just by human eyes but by AI algorithms trained on vast databases. This study highlights the impressive strides made in radiomics, where AI extracts intricate features from medical images, aiding in the detection and classification of urological cancers, including prostate, bladder, and renal tumors. By analyzing subtle patterns invisible to the naked eye, AI can improve diagnostic accuracy, leading to earlier diagnoses and better treatment outcomes. Beyond cancer detection, AI is poised to revolutionize surgical planning and execution. Algorithms trained on surgical videos and data can predict potential complications, optimize surgical routes, and guide surgeons in real time, minimizing invasive procedures and improving patient safety. The study showcases the potential of AI-assisted robotic surgery, where robots equipped with AI software operate with greater precision and efficiency, leading to faster recovery times and reduced risk of complications. But AI’s impact extends beyond diagnosis and surgery. It can also personalize patient care through predictive modeling. By analyzing a patient’s medical history, genetics, and lifestyle factors, AI algorithms can predict their risk of developing urological conditions, allowing for preventative interventions and early detection. The study highlights the use of AI in predicting the likelihood of stone recurrence, guiding personalized follow-up and treatment strategies [52].

Furthermore, AI empowers urologists with decision support tools. Imagine having access to a vast library of medical knowledge and clinical data, instantly accessible at your fingertips. AI-powered systems can analyze this information, providing real-time recommendations and tailoring treatment plans to individual patients based on the latest evidence and best practices. This democratization of knowledge empowers urologists to make informed decisions while ensuring consistent, high-quality care for all patients. The future of AI in urology is brimming with possibilities. The study envisions a future where AI-powered chatbots offer patients 24/7 support and guidance, managing chronic conditions and answering questions. Imagine AI-driven personalized medicine, where drugs are targeted to individual genetic profiles, maximizing treatment efficacy and minimizing side effects.

Crucially, within the realm of AI’s potential, there is significant opportunity for knowledge expansion in utilizing miRNA to distinguish teratoma from necrosis and viable tumors. As technology advances, the synergy between AI and miRNA holds immense promise in elevating the accuracy and effectiveness of diagnostics in urology.

### 6.3. Potential of Clinical Application

The potential of miRNA in clinical applications is indeed substantial. The high sensitivity and specificity of miRNA enable the early detection of disease recurrence following primary treatment. Furthermore, miRNA analysis provides a mechanism for differentiating between viable GCT, teratoma, and fibrosis. This is particularly beneficial in identifying non-seminoma patients with residual masses ≥1 cm following chemotherapy. As a result, this facilitates the judicious avoidance of unnecessary pcRPLND. This strategic approach underscores the pivotal role of miRNA in enhancing patient care and improving treatment outcomes.

However, it is crucial to acknowledge the limitations associated with the utility of miRNA. Many of the studies discussed are based on relatively small sample sizes, which could potentially compromise the generalizability and reliability of their findings. To bolster the robustness of the evidence presented, it would be essential to conduct studies with larger and more diverse cohorts. The retrospective design employed in some studies introduces bias and hinders the establishment of causal relationships, thereby emphasizing the need for prospective investigations. The absence of external validation in certain studies is a significant limitation, as replicating results in independent cohorts is crucial for confirming the accuracy and reliability of identified biomarkers. Lastly, the lack of longitudinal data in the article hinders a comprehensive understanding of dynamic changes in miRNA expression over time and their correlation with disease progression. These considerations are vital for the future development and application of miRNA in clinical practice.

## 7. Conclusions

miRNAs, specifically miR-371 and miR-375, have emerged as promising candidates for the diagnosis and differentiation of TGCTs, with a particular focus on identifying teratomas. These small non-coding RNA molecules play a critical role in gene regulation and have shown potential as biomarkers due to their stability in serum. The studies discussed here shed light on their utility and limitations in the context of TGCTs, especially when it comes to distinguishing teratomas. Teratomas pose a unique challenge in diagnosis, as they exhibit high degrees of differentiation and share characteristics with normal somatic tissues. This complexity necessitates the search for precise circulating biomarkers, and miRNAs have offered intriguing possibilities. Notably, miR-371 has shown promise in detecting both seminoma and non-seminoma cases but falls short in distinguishing between distinct teratoma subtypes. In contrast, miR-375, despite its elevated expression within teratomas, has faced challenges in providing a clear discriminatory power. While it may not stand alone as a reliable teratoma identifier, studies suggest that combining miR-371 and miR-375 could offer a more accurate means of differentiation. The integrated evaluation of these miRNAs, as demonstrated by Nappi et al. and Kremer et al., presents a potential solution with high sensitivity and specificity. However, it is important to acknowledge the limitations of these studies, including sample sizes and the need for external validation. Further research is warranted to refine and validate these miRNA-based approaches for teratoma identification. If successful, these miRNA signatures could revolutionize TGCT diagnosis, reduce overtreatment, and provide more targeted therapeutic strategies, ultimately improving patient outcomes. Teratoma identification, a challenging endeavor, may soon benefit from the precision of miRNA-based biomarkers.

## Figures and Tables

**Table 1 ijms-25-02156-t001:** miR-371 for TGCT primary diagnosis.

Author	Year	Patient (n)	Sensitivity (%)	Specificity (%)	PPV (%)	NPV (%)	AUC
Dieckmann [30]	2019	616	91.8	96.1	97.2	82.7	0.97
Nappi [31]	2019	110	96	100	100	98	0.97
Badia [32]	2021	69	93.1	100	100	73.3	0.98
Sequeira [33]	2022	82	93.6	100	100	96	0.98

Note: PPV = positive predictive value, NPV = negative predictive value, and AUC = area under the curve.

**Table 2 ijms-25-02156-t002:** miR-375 for identifying teratoma.

Author	Year	Number of Patients (Total)	miRNA Type	AUC
Lafin [40]	2021	40	miR-375-3p	0.506
			miR-375-5p	0.556
Moore [41]	2022	22	miR-375	0.62
Belge [42]	2020	45	miR-375-3p	0.524

Note: AUC = area under the curve, miRNA = microRNA.

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
