# Peer review of "MicroRNAs in Testicular Germ Cell Tumors: The Teratoma Challenge"

_ijms, 2024, doi:10.3390/ijms25042156_

Round 1

Reviewer 1 Report

Comments and Suggestions for Authors

The final part of the Abstract seems confused and should be improved. It doesn't explain that the use of single miRNAs was not enough to identify the teratoma presence, but their combination is instead promising. This is clearly described later, but in the Abstract seems to say the opposite.

The physiological biological role of miR-371 and miR-375 should be explained a bit better. 

In the Discussion, the sections 6.1 and 6.2 seems to be added just for volume. It should be indicated the connection of these new possibilities with the miRNAs or with the info already showed in the review.

There are minor typos in rows: 17, 45, 168-169 for example.

Not consistence in naming: miR375, miRNA375 and miR-375 in the same paragraph.

Comments on the Quality of English Language

The article should improve the language and the connections between the sections.

The initial part of the Abstract is hard to read: are the information are correct, but the single sentences are detached and there is no common thread. 

The section about the miR-375 in the rows 193-200, has no connection with the previous discussion about the miR-371.

Author Response

Thank you very much for taking the time to review this manuscript. Please find the detailed responses below and the corresponding revisions in track changes in the re-submitted files.

Point-by-point response to Comments and Suggestions
Comments 1: The final part of the Abstract seems confused and should be improved. It doesn't explain that the use of single miRNAs was not enough to identify the teratoma presence, but their combination is instead promising. This is clearly described later, but in the Abstract seems to say the opposite.
Response 1: Thank you for pointing this out. We agree with this comment. Therefore, we have revised the abstract to make it clearer, more concise, and easier to understand.

Comments 2: The physiological biological role of miR-371 and miR-375 should be explained a bit better.
Response 2: We agree. We have revised and added more information about miR-371 in rows 300-305, and about miR-375 in rows 340-349.

Comments 3: In the Discussion, the sections 6.1 and 6.2 seems to be added just for volume. It should be indicated the connection of these new possibilities with the miRNAs or with the info already showed in the review.
Response 3: Thank you for pointing this out. Currently, there is no ongoing research establishing a connection. However, we are exploring the potential of using new technology, or a combination of mRNA and miRNA, for teratoma detection. If you believe this section is not suitable for inclusion in this article, we would be more than happy to remove it.

Comments 4: There are minor typos in rows: 17, 45, 168-169 for example.
Response 4: We have rechecked the entire article and corrected all the typos.

Comments 5: Not consistence in naming: miR375, miRNA375 and miR-375 in the same paragraph.
Response 5: We agree. We have revised all the terms and abbreviations for consistency.

Response to Comments on the Quality of English Language

Point 1: The article should improve the language and the connections between the sections.
Response 1: We have tried to improve the language and connections between the sections. We have sequenced the explanation of the problem in current clinical practice, introduced miRNA, explained about miR-371 and miR-375, explained the failure of single miRNA in teratoma detection, and then explained the promise of the combination of miRNA.

Point 2: The initial part of the Abstract is hard to read: are the information are correct, but the single sentences are detached and there is no common thread. 
Response 2: Thank you for pointing this out. We agree with this comment. Therefore, we have revised the abstract to make it clearer, more concise, and easier to understand.

Point 3: The section about the miR-375 in the rows 193-200, has no connection with the previous discussion about the miR-371.
Response 3: We agree. We have tried to explain the second miRNA, miR-375. We revised it and made it connected with the previous paragraph.

Reviewer 2 Report

Comments and Suggestions for Authors

Article by Yodkhunnatham entitled MicroRNAs in Testicular Germ Cell Tumors: The Teratoma comprehensively cover clinically important topic in GCTs management. Article is concisely written, conclusions are supported by  data. 

Major points: 

None

Minor points:

Lines 70-83: I suggest to rephrase this paragraph as seminomas are differently treated compare to non-seminomas e.g. there is no role for radiotherapy in the management of non-seminomas. Furthermore, I sugest to add metastasectomy in addition to RPLND after chemotherapy, and this topic are also valid for patients that needed metastasectomy beyond RPLND

Author Response

Thank you very much for taking the time to review this manuscript. Please find the detailed responses below and the corresponding revisions in track changes in the re-submitted files.

Point-by-point response to Comments and Suggestions

Comments 1: I suggest to rephrase this paragraph as seminomas are differently treated compare to non-seminomas e.g. there is no role for radiotherapy in the management of non-seminomas. Furthermore, I sugest to add metastasectomy in addition to RPLND after chemotherapy, and this topic are also valid for patients that needed metastasectomy beyond RPLND
Response 1: Thank you for pointing this out. We agree with your comment. Therefore, we have made revisions according to your suggestion in rows 151-162.

Reviewer 3 Report

Comments and Suggestions for Authors

I have reviewed the article titled "MicroRNAs in Testicular Germ Cell Tumors: The Teratoma Challenge," presenting the abstract on testicular germ cell tumors (TGCT) by Yodkhunnatham et al. This article focuses on the role of microRNAs (miRNAs) as biomarkers in testicular germ cell tumors, with a particular emphasis on the challenge of non-invasive detection of teratomas. While the review addresses several crucial aspects, there are some additional considerations for a more comprehensive review.

In Section 2, the second part lacks references. In Section 4, there is a significant amount of text without proper citation. I believe it is essential to briefly address other potential biomarkers besides miRNAs in Section 4, such as genetic markers, proteins, or other molecules that may be under investigation and could complement the information provided by miRNAs.

It is pertinent to delve deeper into the underlying molecular mechanisms that link miRNAs to the progression of TGCTs. How do miRNAs impact the cellular and molecular processes involved in the development of teratomas and other TGCT subtypes?

In the discussion, more information is needed on the practical implementation of miRNAs in clinical settings, with a special emphasis on teratomas. How can they be effectively integrated into diagnostic and monitoring protocols for patients with TGCTs? What practical challenges are associated with their clinical implementation? Are there specific therapeutic approaches that could target the identified miRNA profiles?

I recommend the inclusion of at least one figure summarizing the potential involvement of miRNAs and teratomas in TGCT. This visual representation would enhance the understanding of the complex interactions discussed in the article.

Overall, addressing these points will contribute to a more thorough and balanced review, providing valuable insights into the current state of research on miRNAs in TGCTs and identifying key areas for future investigations and clinical

Author Response

Thank you very much for taking the time to review this manuscript. Please find the detailed responses below and the corresponding revisions in track changes in the re-submitted files.

Point-by-point response to Comments and Suggestions
Comments 1: In Section 2, the second part lacks references. In Section 4, there is a significant amount of text without proper citation. 
Response 1: Thank you for bringing this to our attention. We have thoroughly rechecked the manuscript and added the necessary citations throughout.

Comments 2: I believe it is essential to briefly address other potential biomarkers besides miRNAs in Section 4, such as genetic markers, proteins, or other molecules that may be under investigation and could complement the information provided by miRNAs.
Response 2: We appreciate your suggestion. We have revised Section 4 and included additional information about other potential biomarkers in rows 513-644.

Comments 3: It is pertinent to delve deeper into the underlying molecular mechanisms that link miRNAs to the progression of TGCTs. How do miRNAs impact the cellular and molecular processes involved in the development of teratomas and other TGCT subtypes?
Response 3: Thank you for your insightful comment. We have expanded our discussion on this topic and included more information in rows 124-138.

Comments 4: In the discussion, more information is needed on the practical implementation of miRNAs in clinical settings, with a special emphasis on teratomas. How can they be effectively integrated into diagnostic and monitoring protocols for patients with TGCTs? What practical challenges are associated with their clinical implementation? Are there specific therapeutic approaches that could target the identified miRNA profiles?
Response 4: We have added a new section (6.3) to discuss the application of miRNAs in clinical practice (rows 773-793).

Comments 5: I recommend the inclusion of at least one figure summarizing the potential involvement of miRNAs and teratomas in TGCT. This visual representation would enhance the understanding of the complex interactions discussed in the article.
Response 5: Thank you for your valuable suggestion. We agree that a visual representation would indeed enhance the understanding of the complex interactions discussed in the article. However, due to the constraints of our current resources and the complexity of the subject matter, we regret to inform you that we are unable to include such a figure at this time. We appreciate your understanding and will certainly consider this for future updates or publications. Your feedback is greatly appreciated.

Reviewer 4 Report

Comments and Suggestions for Authors

This is an interesting work reviewing MicroRNAs as emerging promising biomarkers for TGCTs focusing also on possible outcomes and on the employment of artificial intelligence as an enhanced diagnostic tool.

The review is well-written, flows smoothly and the content covered is thorough.

Specific minor comment:

-       I suggest an improvement in the introduction describing tumor classification, tumor development and available mouse models of germ cell tumors.

(Guida E. et al., MAPK activation drives male and female mouse teratocarcinomas from late primordial germ cells, Journal of Cell Science 2022.

Pierpont T. M. et al., Chemotherapy-induced depletion of OCT4-positive cancer stem cells in a mouse model of malignant testicular cancer. Cell Rep 2017.

Meng, X. et al., Promotion of seminomatous tumors by targeted overexpression of glial cell line-derived neurotrophic factor in mouse testis. Cancer Res 2001)

-       Some sentences lack of references. Authors should check the manuscript to fix the problem.

-  e.g. Line 108 In recent years, microRNAs (miRNAs) have emerged as compelling candidates forserving as biomarkers in the context of TGCTs needs this reference: https://doi.org/10.3390/life11080736

Author Response

Thank you very much for taking the time to review this manuscript. Please find the detailed responses below and the corresponding revisions in track changes in the re-submitted files.

Point-by-point response to Comments and Suggestions
Comments 1: I suggest an improvement in the introduction describing tumor classification, tumor development and available mouse models of germ cell tumors.
(Guida E. et al., MAPK activation drives male and female mouse teratocarcinomas from late primordial germ cells, Journal of Cell Science 2022.
Pierpont T. M. et al., Chemotherapy-induced depletion of OCT4-positive cancer stem cells in a mouse model of malignant testicular cancer. Cell Rep 2017.
Meng, X. et al., Promotion of seminomatous tumors by targeted overexpression of glial cell line-derived neurotrophic factor in mouse testis. Cancer Res 2001)
Response 1: Thank you for pointing this out. We agree with your comment. Therefore, we have revised the text and included these references in rows 124-138.

Comments 2: Some sentences lack of references. Authors should check the manuscript to fix the problem. e.g. Line 108 In recent years, microRNAs (miRNAs) have emerged as compelling candidates forserving as biomarkers in the context of TGCTs needs this reference: https://doi.org/10.3390/life11080736
Response 2: We agree. We have rechecked and corrected the entire manuscript. We also added the reference: https://doi.org/10.3390/life11080736.

Reviewer 5 Report

Comments and Suggestions for Authors

The authors have attempted to investigate the recent progress regarding the utility of miRNA biomarkers in TGCTs, with an emphasis on outstanding issues surrounding noninvasive detection of teratoma.

The study is methodologically well performed and is well written.

What is the significance of this marker in clinical practice?

Do you think there is a place for clinical application of these markers considering that it is a rare disease?

Please provide reference justifying A recent study published in Cancer 2023 offers a beacon of hope, shedding light on a potential solution to this longstanding dilemma. “ Sectio 6.1

Please list the limitations of this study at the end of the discussion section.

Author Response

Thank you very much for taking the time to review this manuscript. Please find the detailed responses below and the corresponding revisions in track changes in the re-submitted files.

Point-by-point response to Comments and Suggestions
Comments 1: What is the significance of this marker in clinical practice?
Do you think there is a place for clinical application of these markers considering that it is a rare disease?
Response 1: Thank you for your insightful question. We have indeed addressed the clinical application of these markers in section 6.3, rows 777-785.

Comments 2: Please provide reference justifying „A recent study published in Cancer 2023 offers a beacon of hope, shedding light on a potential solution to this longstanding dilemma. “ Sectio 6.1
Response 2: We have added the appropriate reference for this statement in row 704.

Comments 3: Please list the limitations of this study at the end of the discussion section.
Response 3: Thank you for pointing this out. We have listed the limitations of our study in rows 785-797.

Round 2

Reviewer 3 Report

Comments and Suggestions for Authors

The authors have made significant changes to the review and pending the editor's decision, I consider it suitable for publication.